# Development of a High-Performance Adhesive with a Microphase, Separation Crosslinking Structure Using Wheat Flour and a Hydroxymethyl Melamine Prepolymer

**DOI:** 10.3390/polym11050893

**Published:** 2019-05-15

**Authors:** Jieyu Zhang, Yi Zhang, Jianzhang Li, Qiang Gao

**Affiliations:** 1Key Laboratory of Wood Material Science and Utilization, Beijing Forestry University, Beijing 100083, China; 15501081026@163.com (J.Z.); luckyyizhang@163.com (Y.Z.); lijzh@bjfu.edu.cn (J.L.); 2Ministry of Education, Beijing Key Laboratory of Wood Science and Engineering, College of Materials Science and Technology, Beijing Forestry University, Beijing 100083, China

**Keywords:** wheat flour (WF), hydroxymethyl melamine prepolymer (HMP), crosslinking network, micro phase separation structure, plywood, wet shear strength

## Abstract

The objective of this study is to use wheat flour (WF) and hydroxymethyl melamine prepolymer (HMP) to develop a low cost, highly water-resistant, starch-based bio-adhesive for plywood fabrication. Three-layer plywood was fabricated using the resultant adhesive, and the wet shear strength of the plywood samples was measured under various conditions. After determining that water resistance was significantly improved with the addition of HMP, we evaluated the physical characteristics of the starch-based adhesive and functional groups and analyzed the thermal stability and fracture surface of the cured adhesive samples. Results showed that by adding 20 wt.% HMP into WF adhesive, the sedimentation volume in the resultant adhesive decreased by 11.3%, indicating that the increase of crosslinking in the structure of the adhesives increased the bond strength, and the wet shear strength of the resultant plywood in 63 °C water improved by 375% when compared with the WF adhesive. After increasing the addition of HMP to 40 wt.%, the wet shear strength of the resultant plywood in 100 °C water changed from 0 MPa to 0.71 MPa, which meets the exterior use plywood requirement. This water resistance and bond strength improvement resulted from (1) HMP reacting with functions in WF and forming a crosslinking structure to prevent moisture intrusion; and (2) HMP self-crosslinking and combining with crosslinked WF to form a microphase separation crosslinking structure, which improved both the crosslinking density and the toughness of the adhesive, and subsequently, the adhesive’s bond performance. In addition, the microphase separation crosslinking structure had better thermostability and created a compact ductile fracture surface, which further improved the bond performance of the adhesive. Thus, using a prepolymer to form a microphase separation crosslinking structure within the adhesive improves the rigidity, toughness, and water resistance of the material in a practical and cost-effective manner.

## 1. Introduction

Starch is a renewable, natural polymer that is inexpensive, non–toxic, biodegradable, and has adhesion and film-forming properties [1,2,3]. In addition, because it is one of the most abundant natural resources, starch-based adhesives demonstrate significant potential for replacing formaldehyde-based adhesives, thereby eliminating formaldehyde emissions from the resultant panels. Starch-based adhesives have a long application history in corrugated case manufacture [4]. However, their low dry bond strength and high water sensitivity limits their application in the wood panel fabrication industry [5]. The starch-based adhesive’s poor bond performance is attributed to the low reactivity and high hydrophilicity of the hydroxyl groups in the starch [6]. In recent years, many researchers have attempted to modify the starch in order to improve the bond performance and/or water resistance of the resultant adhesive. Such attempts include, but are not limited to oxidized modification [7], enzymatic modification [8,9], esterified modification [10,11,12,13], and grafting modification [14,15]. The adhesives produced from oxidized and enzymatic modification underwent a complicated synthesis process and still depicted a low bond strength and poor water resistance. While esterification modification greatly improves the bond strength and water resistance of the resultant starch-based adhesive, this modification process leaves toxic isocyanate residual. Furthermore, the products have a low pot life and issues with sticking to the hot press plate, which limits their application. Grafting modification, in which a carboxyl, vinyl, and/or other high reactivity functional group is grafted onto the starch molecule, facilitates the formation of a crosslinking structure that improves bond strength and water resistance of the resultant adhesive [16]. However, the grafting process is complex and costly.

Based on our previous research [17,18,19,20,21,22], using a high reactivity crosslinker to create a crosslinked structure in a starch-based adhesive is a simple and effective way of improving the bond performance and reducing the adhesive cost. In this study, laboratory synthetic low cost hydroxymethyl melamine prepolymer (HMP) was synthesized and mixed with wheat flour (WF) to obtain a bio-based wood adhesive. The synthesized adhesive was then applied to three-ply plywood specimens. The wet shear strength of those samples was measured in accordance with the Chinese National Standards (GB/T 17657-1999) and found to be significantly improved. Finally, the physical characteristics, functional groups, and thermo-stability of the cured adhesive were characterized and analyzed to better understand the water resistance mechanism.

## 2. Materials and Methods

### 2.1. Materials

Wheat flour (64% carbohydrate, 20% protein, 13% moisture, 3% fat and ash) was obtained from Beijing Ancient Ship Co., Ltd (Beijing, China) and milled into 200-mesh flour. Formaldehyde, melamine and sucrose were obtained from Tianjin Chemical Reagent Co. (Tianjin, China). HMPs were synthesized in the laboratory. Sodium hydroxide (NaOH), purchased from Tianjin Chemical Reagent Co. (Tianjin, China), was of analytical grade. Poplar veneer (40 cm × 40 cm × 1.5 cm, 8% moisture content) was provided from Wen’an, Hebei, China. Deionized water was used in all experiments.

### 2.2. Preparation of Hydroxymethyl Melamine Prepolymer (HMP)

To prepare HMP, 210 g of Formaldehyde, 100 g distilled water, and 8 g sucrose were placed into a three-necked bottle (i.e., reactor), after which the pH was adjusted to 9.0 with NaOH 20%. Next, 110 g of melamine was added to the mixture, which was heated to 85 °C over 20–30 min. The pH and temperature were maintained at 8.5–9.0 and 85 °C, respectively, for 1 h. Following this, 40 g of melamine was again added to the mixture, and the reaction was permitted to continue at ≥ 85 °C with a pH ≥ 9.0 until a clear fog appeared. Finally, the mixture was cooled to 40 °C and the pH was adjusted to 9.0. Free formaldehyde was removed by a vacuum distillation process, then water was added to adjust the solid content to 50% ± 2%.

### 2.3. Preparation of Adhesives

To prepare the various adhesive samples, WF was added to deionized water and stirred for 10 min at 20 °C. Then, HMP was sequentially added and the mixture was again stirred for 10 min at 20 °C. The adhesive formulations are depicted in Table 1.

### 2.4. Preparation of the Plywood Sample

Three-ply plywood samples were each coated with 180 g/m^2^ of adhesives and hot-pressed for 6 min at 120 °C and 1.0 MPa of pressure. An uncoated veneer was stacked between two adhesive-coated veneers with the grain directions of two adjacent veneers perpendicular to each other [23]. After hot pressing, the plywood samples were stored under ambient conditions for at least 12 h before being tested.

### 2.5. Sedimentation Volume Evaluation

First, each sample was accurately weighed to 0.6 g and then placed into an 80 mL beaker filled with 30 mL of distilled water to make a 2% solution. Next, the solution was placed into a water bath at 85 °C for 2 min, then cooled down to room temperature. Once at room temperature, the solution was centrifuged for 3 min at a rotation speed of 4000 r/min. Finally, the supernatant volume was measured with a measuring cylinder. Three measurements were performed for each sample and the average value was reported with standard deviations [24]. The sedimentation volume is defined by Equation (1):*S* (mL) = 10 − *V* (mL)(1)
where *S* is the sedimentation volume and *V* is the supernatant liquid product. 

### 2.6. Apparent Viscosity Measurement

The viscosity of all the samples was measured on a rheometer, operating with a 20 mm diameter parallel plate, and a distance of 1 mm. The experiments were conducted under a steady shear flow at 20 °C with a spinning rate of 2 rpm. All measurements were performed three times and the average value was reported [25].

### 2.7. Attenuated Total Reflectance–Fourier Transforms Infrared (ATR–FTIR) Spectroscopy Investigation

The samples were placed in an oven at 120 ± 2 °C until a constant weight was obtained. The dried adhesives were ground into 200 mesh powder. ATR-FTIR spectra of cured adhesives were carried out on a Nicolet 6700 FTIR Spectrometer (Thermo Scientific, Pittsburgh, PA, USA) [26]. A total of 32 scans were performed at 4 cm^−1^ resolution with scan width of 4000–600 cm^−1^.

### 2.8. Wet Shear Strength Measurement

The tensile shear strength was tested according to the China National Standards (GB/T 17657–1999) to determine if the plywood samples meet the demand of the interior use [27]. Twelve bonded plywood specimens (2.5 × 10 cm) were cut from two plywood panels and their dry bond strength was tested using a tensile machine at an operating speed of 20.0 mm·min^−1^. For the wet shear strength measurements, plywood panels were tested after being submerged in water at 63 ± 2 °C or boiled at 100 ± 2 °C for 3 h and then dried at room temperature for 10 min before tension testing. The force required to break the resultant plywood was measured under the same testing conditions. The tensile shear strength was calculated by Equation (2):(2)$$\mathrm{Bonding}\;\mathrm{strength}\;\left(\mathrm{MPa}\right)=\frac{\mathrm{Tension}\;\mathrm{Force}\;\left(N\right)}{\mathrm{Gluing}\;\mathrm{area}\;\left(m^{2}\right)}$$

### 2.9. Thermogravimetric (TG) Measurement

The samples were placed in an oven at 120 °C ± 2 °C until a constant weight was obtained and ground into a powder. The stabilities of the cured adhesives were tested by using a TGA instrument (TA Q50, WATERS Company, USA). Samples (5 mg in a platinum cup) were heated from room temperature to 600 °C at a heat rate of 10 °C min^−1^ under nitrogen environment while recording the weight changes in the samples [28].

### 2.10. Scanning Electron Microscopy (SEM)

After the adhesive samples were completely cured at 120 ± 2 °C, the adhesives were broken and the fracture surface of the samples were sputter-coated with 10 nm of Au/Pd film using a Q150T S Turbo-Pumped Sputter Coater/Carbon Coater (Quorum Technologies Ltd., East Sussex, UK) to ensure sufficient conductivity, then characterized by a Hitachi S-4800 emission scanning electron microscope (Hitachi Scientific Instruments, Tokyo, Japan) under 100 and 3000 × magnification [29].

## 3. Results and Discussion

### 3.1. HMP Analysis

The reaction between melamine and formaldehyde is complete and non-reversible; thus, formaldehyde emissions are not an issue of concern [30]. Figure 1a shows the FTIR spectra and the reaction schematic of HMP. Note that HMP shows a peak at 2950 cm^−1^ that is not present in the melamine spectra; this is attributed to C–H stretching vibrations. In addition, the following are also observed in the HMP spectra: An absorption peak at 3325 cm^−1^, which represents amino (–NH_2_) and amino (–NH–) vibrations [31]; a methylene C–H bending vibration at 1485 cm^−1^ [32]; and a secondary amine C–N stretching at 1157 cm^−1^ [33]. All of these findings indicate that melamine reacted with formaldehyde to develop methylol melamine and these methylol melamine further formed a prepolymer with –NH–CH_2_–NH– linkage by a self-condensation reaction (Figure 1b). Finally, the characteristic 1, 3, 5-triazine ring peak appeared at 810 cm^−1^, demonstrating that the melamine structure was not destroyed in the reaction [34].

### 3.2. ATR-FTIR Analysis

The FTIR spectra of the adhesive samples are presented in Figure 2. The FTIR spectrum of the WF adhesive depicts three characteristic absorption bands at 2925 cm^−1^, 1644 cm^−1^, and 1160–1010 cm^−1^ are attributed to C–H stretching, O–H bending vibration, and anhydroglucose ring O–C stretch, respectively. The infrared bands at approximately 3305–3325 cm^−1^ are assigned to hydrogen bonded O–H stretching vibration and N–H bond vibration [35,36]. As the HMP concentration increased, two evident changes occurred: (1) The typical absorption peak at 1644 cm^−1^, which represents the –OH group on the WF molecules, decreased significantly or, in some cases, completely disappeared; and (2) the O–H stretching peak at 3305 cm^−1^ also decreased. Both results together indicate that the –OH group was consumed, which signifies that either a methylene bond or ether bond formed between the WF and HMP molecules.

After bonding between WF and HMP was complete, HMP’s characteristic 1, 3, 5–triazine ring peak at 810 cm^−1^ appeared, as did the methylene C–H peak at 1484 cm^−1^, which shows that HMP was present in the resultant adhesives. Furthermore, the HMP stretching vibration peak at 810 cm^−1^ in the WF/HMP-20 sample was stronger than that of the WF/HMP-10 sample, suggesting that the HMP was well dispersed in the adhesive. In addition, the peak at 1484 cm^−1^ was due to methylene C–H bending vibration of the pure HMP. By comparison, when the peak at 810 cm^−1^ corresponded to the 1,3,5–triazine ring, the new –CH_2_– group peak at 1495 cm^−1^ gradually gained intensity, indicating generation of a methylene bond between WF and HMP molecules, which coincides with a decrease in the number of –OH groups associated with WF molecules. Meanwhile, the peak at 1204 cm^−1^ appeared corresponding to C–O bending, implying that the WF and HMP was linked by ether bonds [34,37]. The increase of the –CH_2_– group peak could also have resulted from HMP self-polycondensation, which increased the cross-linking density of the resultant adhesives. The schematic of crosslinking reaction in the adhesives was showed in Figure 3.

### 3.3. Sedimentation Volume Analysis

Starch swells in the water and this process can be measured by detecting starch sedimentation volume. When starch was crosslinked, the swelling property was increased, resulting in the starch sedimentation volume reducing. Therefore, the sedimentation volumes could be used to evaluate the crosslinking structure in the adhesive and WF [24,38]. The sedimentation volumes of all the adhesive samples are shown in Table 2. The HMP and WF featured a sedimentation volume of –16 mL and –15.5 mL, respectively, whereas a lower sedimentation volume was observed for the combined adhesives. With the addition of 10 wt.% HMP, the sedimentation volume of the resultant adhesive decreased 6.5% in comparison to pure WF, denoting a 6.5% increase in the crosslinking structure. As the HMP concentration increased, the sedimentation volume of WF/HMP adhesives reduced to −18.5 mL in WF/HMP-40 adhesive; a 19.4% decrease from the WF adhesive. Thus, the crosslinking structure increased with increasing HMP content. The increase might have resulted from (1) the reaction between the HMP and WF molecules or HMP self-polycondensation forming a compact and homogeneous structure, which reduced the number of the hydrophilic groups and improved the hydrolytic stability of the cured adhesive; (2) the formation of a dense structure, which reduced the WF molecules solubility and improved the water resistant of the adhesives (consistent with SEM results); and/or (3) HMP self-polycondensation interweaved with WF molecules to form an HMP-based crosslinking structure, which improved the amount of crosslinking structures in WF molecules and prevented moisture intrusion.

### 3.4. Apparent Viscosity Measurement

The apparent viscosity largely affects the adhesive’s performance. Excessive viscosity causes a hard, inflexible coating on the wood surface; and overly low viscosity results in over-penetration issues [39]. Both extremes cause bond strength reduction in the resultant plywood. According to the GB/T2794-1995, a suitable viscosity for plywood adhesive ranges from 5000 to 25,000 mPa·s [22]. As shown in Table 3, the initial viscosity of the WF adhesive was 2283 mPa·s, but this value increased to 5258 mPa·s with the introduction of 10 wt.% HMP (WF/HMP-10). This improvement was due to the swelling effect of WF in the adhesive system, which increased the hydrogen bond interaction between HMP molecules and dispersed WF, causing the viscosity in the adhesive to gradually increase. A further increase of HMP to 20 wt.% (WF/HMP-20) generated a viscosity of 8212 mPa·s and the viscosity of the WF/HMP-40 sample increased by 1518.9% when compared to that of the pure WF adhesive. The consistent increase in viscosity is ascribed to the friction between molecules resulting from the formation of more hydrogen-bonds between the HMP and WF. Moreover, because the pH of the HMP was 9.0, adding more HMP improved the pH of the adhesive. This, in turn, denatured the WF molecules, which further increased the viscosity of the adhesive. Due to the addition of HMP to the WF adhesive, the viscosity of the modified adhesives ranged from 5258 to 13,220 mPa·s, which satisfied the plywood production requirement.

### 3.5. TG Analysis

Figure 4 shows the thermo gravimetric (TG) and derivative thermo gravimetric (DTG) curves of the adhesive samples. The thermal degradation of the adhesive is divided into three stages: The first stage (I), observed between 0 and 140 °C, was primarily associated with the evaporation of residual moisture. The second stage (II), which ranged from 140 to 250 °C, was associated with the thermal degradation of the unstable chemical bonds and the small molecules in the adhesives. The third stage (III) ranged from 245 to 350 °C and was the skeleton structure degradation stage, which resulted from the dehydration of the polymer chains [40]. The degradation peak of HMP in the DTG curve was observed at 400 °C, while the degradation peak of the WF adhesive was observed at 300 °C. As HMP was added, the third stage degradation peaks of the adhesives increased from 289 to 310 °C, indicating that the resultant adhesive had a better thermal stability. Coincidentally, the third stage DTG curve of the resultant adhesives was slower than both the pure HMP and WF adhesives, which signified that this compound was more thermally stable than either HMP or WF alone. Essentially, the addition of HMP facilitated the formation of a denser crosslinking structure between the HMP and WF molecules, which also effectively improved the water resistance of the adhesive. Finally, the weight loss of the WF/HMP-10 and WF/HMP-40 adhesive was lower than that of the pure WF adhesive by 7.4% and 28.3%, respectively. These results confirm that the crosslinking structure has higher thermal stability and thus improves the thermostability of the adhesive.

### 3.6. Plywood Shear Strength Measurement

The shear strength of plywood bonded with different sample adhesives is shown in Figure 5. The pure HMP has low dry shear strength and exhibits no strength in wet conditions, which is attributed to the fact that a low concentration of pure HMP molecules penetrated into the sheet. The dry shear strength of plywood bonded with the WF adhesive was 1.33 MPa and the wet shear strength, tested in 63 °C water, was 0.16 MPa. Thus, the resultant plywood has a low water resistance and does not meet the requirement for interior use of plywood. By adding 10 wt.% HMP, the dry shear strength of the plywood bonded with WF/HMP-10 adhesive increased by 20.3% relative to that of the WF adhesive. However, no evident change was observed in the dry shear strength of the plywood from increasing the HMP concentration further. Moreover, sample WF/HMP-30 demonstrated 100% wood failure, implying that the bond strength of the adhesive layer was higher than that of the wood strength. In contrast, the wet shear strength of plywood in 63 °C water increased with the addition of HMP. When using sample WF/HMP-20, the wet shear strength of plywood increased to 0.76 MPa, meeting the interior use plywood requirements (≥ 0.7 MPa). Application of sample WF/HMP-40 increased the wet shear strength of the plywood to 1.73 MPa, a 981% increase over the WF adhesive. In the 100 °C water, the plywood bonded by WF adhesive showed no intensity and the wet shear strength increased to only 0.05 MPa with use of WF/HMP-10. Application of sample WF/HMP-40 yielded plywood wet shear strength of 0.71 MPa, which meets the exterior-use plywood requirement. This increasing wet shear strength is due to the HMP reacting with active groups in the WF molecule and forming a cross-linked structure. In addition, HMP underwent self-polycondensation and formed a denser cross-linked network with carbohydrate in the adhesive. Both processes improved the water resistance of the adhesive. The cross-linking scheme is illustrated in Figure 3. 

### 3.7. SEM Analysis

Figure 6 shows the fracture surface morphology of various cured adhesive samples examined by SEM. The cured WF adhesive’s showed a disordered surface, characterized by cracks and holes, which is attributed to water evaporation in the adhesive during the curing process [41]. As no chemical reactions took place during the curing, the bond strength of the WF adhesive came from the physical interlocking of the adhesive molecules and hydrogen bonds formed by the hydroxyl groups on the starch molecules. Thus, the WF adhesive was easily intruded and broken down by water and moisture. After the incorporation of HMP, a more compact, organized surface with fewer cracks was observed in the cured WF/HMP-10 adhesive; this trend continued with the fracture surface of sample WF/HMP-20, which was smoother and more compact than the WF/HMP-10 adhesive fracture surface. The compact surface is a by-product of the strong cross-linked network formed between the HMP and the WF molecules and effectively prevents moisture intrusion thereby improving the water resistance of the adhesive.

Wrinkles were observed on the fracture surface of the WF/HMP-20 adhesive, presenting a toughness fracture surface which was the typical breakage of a toughness adhesive. More wrinkles were observed on the surface of the WF/HMP-40 adhesive, indicating that the adhesive’s toughness was increasing. Thus, the cured adhesive first presents an increase in brittleness, but then the toughness of the adhesive increased in parallel with increasing HMP concentration. This phenomenon is explained by the notion that the reaction between HMP and WF plays a dominate role in the adhesive curing process at a low addition of HMP (Figure 7a). When the addition of HMP increases to ≥ 20 wt.%, HMP molecules have more chance to meet with each other and self-crosslink to form an HMP based crosslinking structure. This HMP based crosslinking structure (hard phase), is penetrated with the WF (soft phase) to form an interpenetrating network structure. These two different phases are connected by the reaction between hydroxymethyl and hydroxyl groups in the adhesive, which form a microphase separation system in the cured adhesive, and effectively improve its toughness (Figure 7b). This elevated adhesive toughness balanced the interior force of the resultant plywood, which improved the bond performance and impacted the water resistance of the adhesive. This conclusion is in agreement with the results of plywood shear strength and the sedimentation volume measurements. The crosslinking process is illustrated in Figure 7.

## 4. Conclusions

The HMP effectively improved the water resistance of the WF adhesive as well as the wet shear strength of the resultant plywood. With the addition of 20 wt.% HMP, the adhesive depicted an 11.3% increase in the amount of crosslinking, a 259.7% increase in viscosity, and a 375% increase in the wet shear strength of plywood submerged in 63 °C water; when compared to the WF adhesive. Further increasing the HMP concentration to 40 wt.%, resulted in a 19.4% decrease in the sedimentation volume indicating the increasing amount of adhesive crosslinking and the wet shear strength of the resultant plywood in 63 °C and 100 °C water were improved to 1.73 MPa and 0.71 MPa, respectively. This significant improvement in adhesive’s bond performance was attributed to: (1) At a low addition of HMP, the reaction between HMP and WF played a dominate role in the adhesive curing process, which created a crosslinked network and improved the water resistance of the adhesive; (2) As the HMP concentration increased, HMP molecules self-crosslinked to form an HMP based crosslinking structure, which penetrated with WF to form a micro phase separation crosslinking structure in the cured adhesive, which effectively both improved the crosslinking density and toughness of the adhesive. This microphase separation crosslinking structure improved the adhesive thermostability, created a compact ductile fracture surface to prevent moisture intrusion, and balanced the interior force of the resultant plywood, which further improved the bond performance of the adhesive. The viscosity of the resultant adhesive ranged from 5258 to 13,220 mPa·s, which is acceptable for the plywood manufacturing requirement.

The use of prepolymers to create microphase separation crosslinking structures improves both the rigidity and the toughness of the material, and can be applied to developing high-performance films, composites, hydrogels, and the other materials.

## Figures and Tables

**Figure 1 polymers-11-00893-f001:**
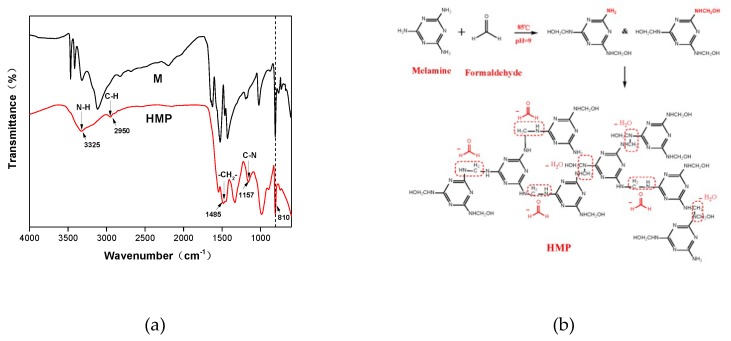
FTIR spectra of HMP (**a**) and the reaction schematic of HMP (**b**).

**Figure 2 polymers-11-00893-f002:**
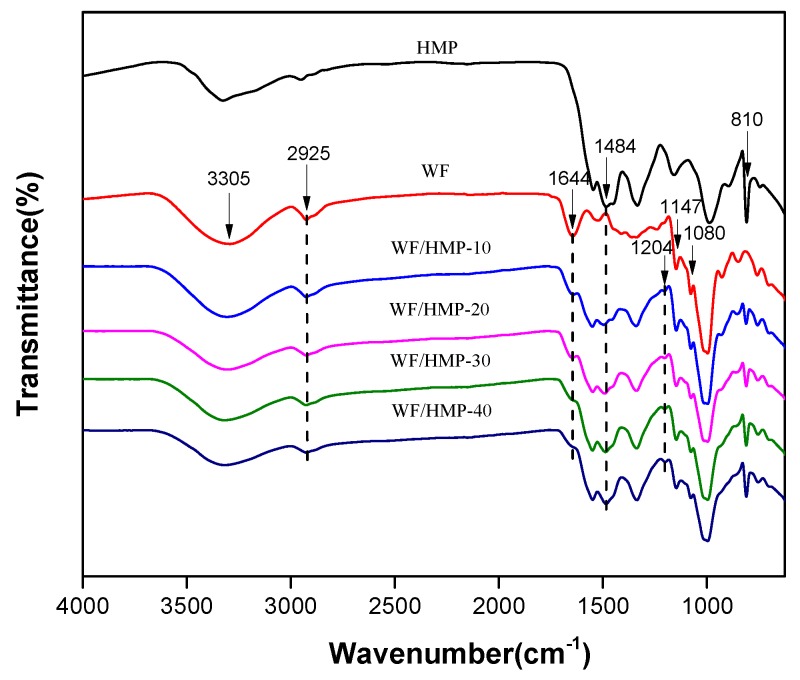
FTIR spectra of the different adhesive samples.

**Figure 3 polymers-11-00893-f003:**
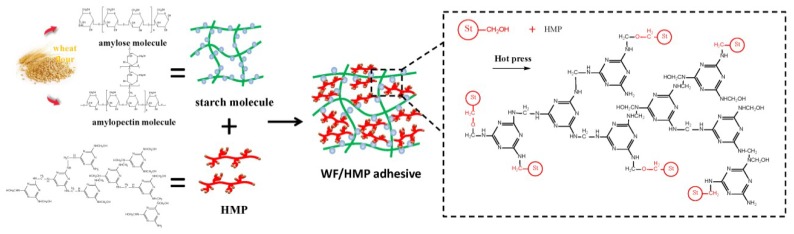
Schematic of crosslinking reaction in the adhesives.

**Figure 4 polymers-11-00893-f004:**
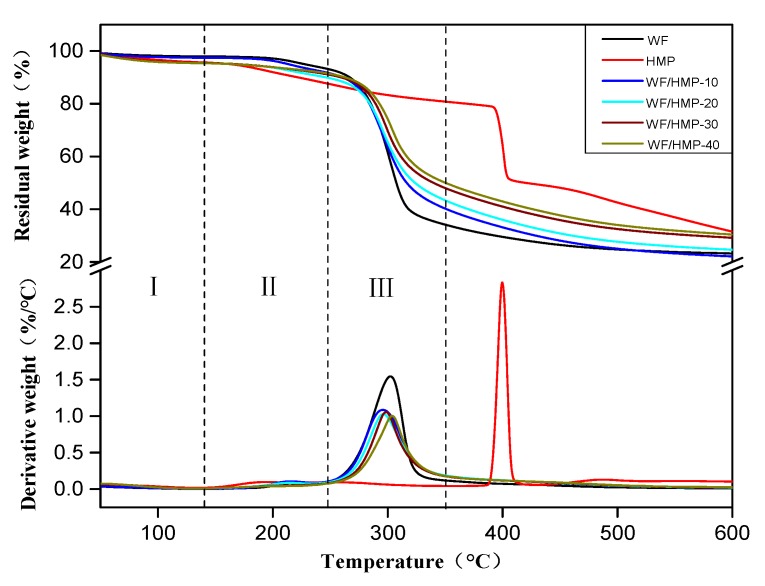
Thermo gravimetric (TG) and derivative thermo gravimetric (DTG) of the different adhesive samples.

**Figure 5 polymers-11-00893-f005:**
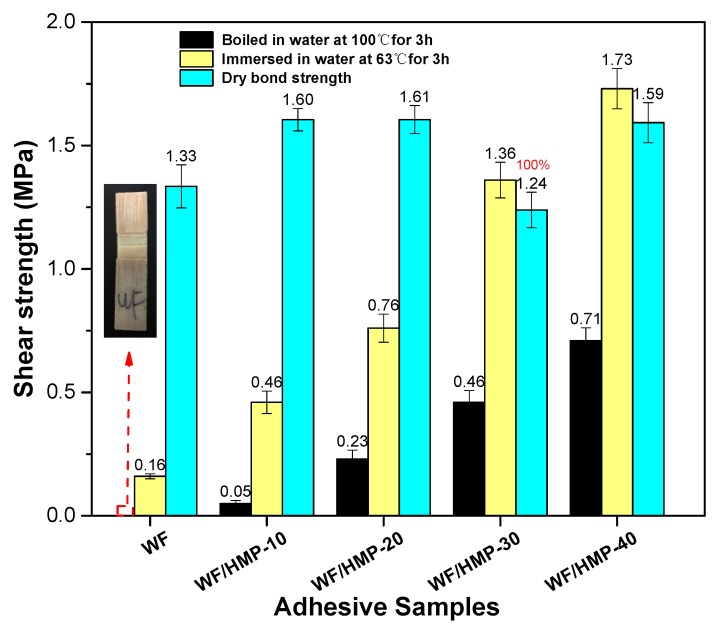
The wet shear strength of the different adhesive samples.

**Figure 6 polymers-11-00893-f006:**
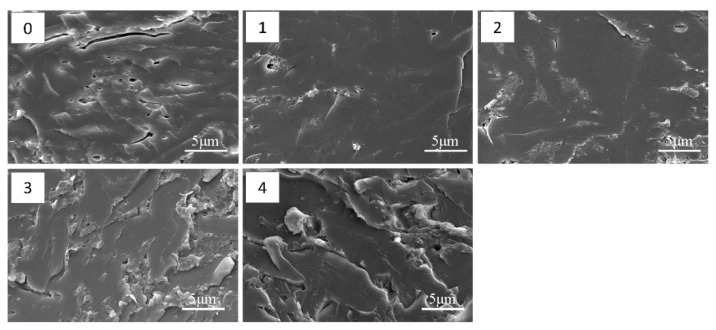
The fracture surface micrograph of different adhesive samples: 0: WF adhesive, 1: WF/HMP-10 adhesive, 2: WF/HMP-20 adhesive, 3: WF/HMP-30 adhesive, 4: WF/HMP-40 adhesive.

**Figure 7 polymers-11-00893-f007:**
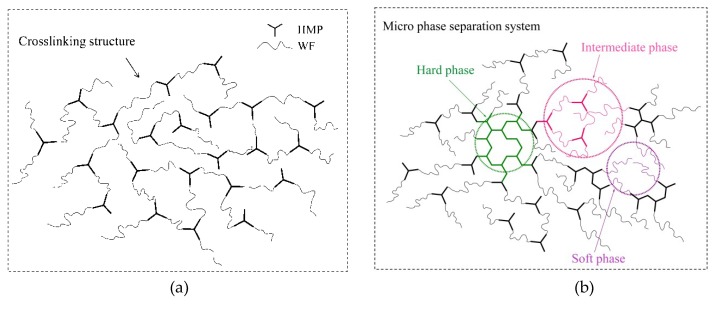
The crosslinking reaction of HMP/WF adhesives: (**a**) Crosslinking structure formed by WF and HMP, (**b**) Micro phase separation structure form by WF and HMP.

**Table 1 polymers-11-00893-t001:** Different wheat flour (WF)/hydroxymethyl melamine prepolymer (HMP) adhesive formulations.

Sample		Adhesive Formulation		
		Wheat Flour (g)	Deionized Water (g)	HMP (g)
0	HMP adhesive	0	0	100
1	WF adhesive	35	65	–
2	WF/HMP–10 adhesive	35	55	10
3	WF/ HMP–20 adhesive	35	45	20
4	WF/ HMP–30 adhesive	35	35	30
5	WF/ HMP–40 adhesive	35	25	40

**Table 2 polymers-11-00893-t002:** The sedimentation volume of different adhesives: 0: HMP adhesive, 1: WF adhesive, 2: WF/HMP-10 adhesive, 3: WF/HMP-20 adhesive, 4: WF/HMP-30 adhesive, 5: WF/HMP-40 adhesive.

**Sample**	0	1	2	3	4	5
**Sedimentation volume (mL)**	−16 ± 0.02 ^a^	−15.5 ± 0.03 ^a^	−16.5 ± 0.05 ^a^	−17.25 ± 0.02 ^a^	−18 ± 0.06 ^a^	−18.5 ± 0.07 ^a^

^a^ Represent standard deviations.

**Table 3 polymers-11-00893-t003:** The apparent viscosity of different adhesives: 0: WF adhesive, 1: WF/HMP-10 adhesive, 2: WF/HMP-20 adhesive, 3: WF/HMP-30 adhesive, 4: WF/HMP-40 adhesive.

**Sample**	0	1	2	3	4
**Initial viscosity (mPa·s)**	2283	5258	8212	13,220	36,960

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
