# Peer review of "Development of a High-Performance Adhesive with a Microphase, Separation Crosslinking Structure Using Wheat Flour and a Hydroxymethyl Melamine Prepolymer"

_polymers, 2019, doi:10.3390/polym11050893_

Round 1

Reviewer 1 Report

I have only one remark:

the authors selected to characterize the cross-linked structure of polymers by the degree of crosslinking. However, the equation No.1 in the subchapter No. 2.5 defines the sedimentation volume, but not the degree of crosslinking. In such case, the authors should give the method of calculation of the degree of crosslinking according to the sedimentation volume.

Author Response

Thank you for your letter and the reviewers’ comments concerning our manuscript entitled “Development of a High-Performance Adhesive with a Microphase, Separation Crosslinking Structure using Wheat Flour and a Hydroxymethyl Melamine Prepolymer” (Manuscript ID: polymers-500795). Those comments are valuable and very helpful for revising and improving our paper, as well as the important guiding significance to our researches. We have studied comments carefully and made correction, and the whole manuscript has been double checked. This manuscript was also polished by a commercial proof-reading company that we hope to meet with approval. Revised portion are marked in red in the Main Article. The main corrections in the paper and the responds to the reviewer’s comments are as flowing:

Reviewer 1:

Point 1: I have only one remark:

the authors selected to characterize the cross-linked structure of polymers by the degree of crosslinking. However, the equation No.1 in the subchapter No. 2.5 defines the sedimentation volume, but not the degree of crosslinking. In such case, the authors should give the method of calculation of the degree of crosslinking according to the sedimentation volume.

Response 1:

The comment is right. We are measuring the “Sedimentation volume Evaluation” of the adhesive and WF. Because the cross-linking structure in starch is low and difficult to directly measure. Sedimentation volume was usually used to characterize the crosslinking structure of starch indirectly according to the literature [1-2]. Starch swells in the water and this process can be measured by detecting starch sedimentation volume. When starch was crosslinked, the swelling property was increased, resulting in the starch sedimentation volume reducing. It is a way to measure the crosslinking structure in the WF adhesive.

1.          Hamdi, G.; Ponchel, G. Enzymatic Degradation of Epichlorohydrin Crosslinked Starch               Microspheres by α-Amylase. [J].1999.1916(1996):1867-1875.

2.        Liu Yawei. Starch production and its deep processing technology [M]. China Light                     Industry Press, 2001.

Also, there is not equation to calculate the degree of crosslinking from Sedimentation volume. So that, in order to avoid misunderstanding, the title of chapter No.2.5 has been changed from “Degree of crosslinking Evaluation” to Sedimentation volume Evaluation”. And the corresponding representation in the chapter No.3.3, abstract and conclusions of the manuscript has been changed and explained.

We appreciate for Editors/Reviewers’ warm work earnestly, and hope that the corrections will meet with approval. Once again, thank you very much for your comments and suggestions. We look forward to your information about my revised papers and thank you for your good comments.  

Yours sincerely,

Jieyu Zhang

Reviewer 2 Report

The authors have done a respectable job revising the paper. I commend their efforts.

Author Response

Thank you for your letter and the reviewers’ comments concerning our manuscript entitled “Development of a High-Performance Adhesive with a Microphase, Separation Crosslinking Structure using Wheat Flour and a Hydroxymethyl Melamine Prepolymer” (Manuscript ID: polymers-500795). Those comments are valuable and very helpful for revising and improving our paper, as well as the important guiding significance to our researches. We have studied comments carefully and made correction, and the whole manuscript has been double checked. This manuscript was also polished by a commercial proof-reading company that we hope to meet with approval. Revised portion are marked in red in the Main Article. The main corrections in the paper and the responds to the reviewer’s comments are as flowing:

Reviewer 2:

Point 1:

The authors have done a respectable job revising the paper. I commend their efforts.

Response 1:

Thank you very much for your recognition of our work.

We appreciate for Editors/Reviewers’ warm work earnestly, and hope that the corrections will meet with approval. Once again, thank you very much for your comments and suggestions. We look forward to your information about my revised papers and thank you for your good comments.  

Yours sincerely,

Jieyu Zhang
